# Development of the petaloid bracts of a paleoherb species, *Saururus chinensis*

Yin-He Zhao[1]*, Xue-Mei Zhang[1], De-Zhu Li[2]*

**1** College of Agronomy and Biotechnology, Yunnan Agricultural University, Kunming, Yunnan, China,
**2** Germplasm Bank of Wild Species, Kunming Institute of Botany, Chinese Academy of Sciences, Kunming, Yunnan, China

* yhzhao808@163.com (YHZ); dzl@mail.kib.ac.cn (DZL)

**Data Availability Statement:** The data are available through GenBank with the accession numbers MZ044086-MZ044244, SRR14085885, SRR14085884 and SRR14085883.

## Abstract

*Saururus chinensis* is a core member of Saururaceae, an ancient, perianthless (lacking petals or sepals) family of the magnoliids in the Mesangiospermae, which is important for understanding the origin and evolution of early flowers due to its unusual floral composition and petaloid bracts. To compare their transcriptomes, RNA-seq abundance analysis identified 43,463 genes that were found to be differentially expressed in *S. chinensis* bracts. Of these, 5,797 showed significant differential expression, of which 1,770 were up-regulated and 4,027 down-regulated in green compared to white bracts. The expression profiles were also compared using cDNA microarrays, which identified 166 additional differentially expressed genes. Subsequently, qRT-PCR was used to verify and extend the cDNA microarray results, showing that the A and B class MADS-box genes were up-regulated in the white bracts. Phylogenetic analysis was performed on putative *S. chinensis* A and B-class of MADS-box genes to infer evolutionary relationships within the A and B-class of MADS-box gene family. In addition, nature selection and protein interactions of B class MADS-box proteins were inferred that B-class genes free from evolutionary pressures. The results indicate that petaloid bracts display anatomical and gene expression features normally associated with petals, as found in petaloid bracts of other species, and support an evolutionarily conserved developmental program for petaloid bracts.

## Introduction

The angiosperm perianth leave perennial always continues to be a focus of evo-devo studies, due to its remarkable diversity across angiosperm species and the likely convergent evolution of its parts [1]. Generally, flowers comprise several sterile laminar structures of the perianth encircling male and female reproductive organs. While some flowers have undifferentiated perianths (tepals), the majority of perianths comprise two different whorls of leaf, i.e., the outer green sepals and the inner, showy petals [2–4]. According to morphological and phylogenetic evidence, petals have evolved multiple times, arising as modifications of stamen-like structures (andropetaloidy) in some lineages, and modifications of bracts or leaves (bracteopetaloidy) in others [5, 6]. The petal organs tend to be large, showy, and either white (due to the presence of leucoplasts) or pigmented (due to the presence of chromoplasts). Petal epidermal cells tend to be conical or elongate, at least on the adaxial face [7].

**Funding:** This work was supported by the National Natural Science Foundation of China [31760059 and 31460053]; The Ministry of Science and Technology of China [2004DKA30430].

**Competing interests:** The authors declare that they have no competing interests.

Research into evolutionary developmental genetics has led to useful models for the genetic basis of perianth development. Foremost is the ABCE model, by which four classes of transcription factors combine to specify each of the four typical floral organs as modified leaves [8–10]. In the ABCE model, class A transcription factors specify the identity of the first and second whorl organs, but combine with B-class factors only in the second whorl organs to form the petals. The B-class genes are also expressed in the third whorl organs, where they combine with C genes to specify androecium development. In the fourth whorl, expression of C-class genes alone leads to gynoecium development [11–14]. E-class genes are expressed in all four whorls, combining with the A, B, and C transcription factors to aid in organ identity specification [15].

Petaloid bracts are found in many other species, such as poinsettia *(Euphorbia pulcherrima)*, the dove tree *(Davidia involucrata)*, and the flowering dogwood, *(Cornus florida)*. The effect of these showy bracts in pollinator attraction is analogous to that of true petals, and their petaloid characteristics probably result from expression of petal development genes in the bracts [5, 16, 17]. Another model, the "fading borders" model, suggested that homologs of B-function MADS-box genes are more broadly expressed across the floral meristem in basal lineages [18]. For example, ectopic expression of the B-class genes *AP3* and *PI* in *Arabidopsis* first-whorl organs results in petaloid organs in place of sepals [5, 19]. The evolution of B-gene function is complex, however, since B-class gene expression has also been found in gymnosperm androecia, and in other non-petal angiosperm organs such as the staminodes [20]. Evidence of expression from orthologs of *Arabidopsis* B-class was reported in the petaloid involucral bracts (bracts that appear in a whorl subtending an inflorescence) of the dove tree *(Davidia involucrata)* by Vekemans et al. [21], with two B-class genes detected early in bract development in this species. The expression of the *CorPI-B*, *CorPI-A* and *CorAP3* genes could be detected in the developing bracts of *Cornus florida*. The detection of B-class genes in the maturing bracts in *Cornus* suggested that the mechanism for heterotopic petaloidy may exist in the *Cornus* lineage [22].

The perianthless paleoherb *S. chinensis* provides a unique opportunity to study the genetics of floral development in a magnoliid of the Mesangiospermae, whose bracts are at least analogs of, and perhaps are homologous to, petals in other angiosperm species. In this work, we compared the morphologies and physiological indices of petaloid and non-petaloid (green) bracts. We then constructed cDNA libraries from the transcriptomes of these two types of bract, assigned the *S. chinensis* genes to functional categories, and measured transcript abundance. We further compared expression profiles of tissue sections of bracts that were partially petaloid; that is, for bracts that were half white and half green, we extracted nucleic acids from the white and green portions separately, both for green bracts transitioning to white, and for white bracts reverting to green coloration following floral senescence. We then validated the results for A and B class genes and four transcription factors by qRT-PCR. Finally, we used phylogenetic analysis to compare *S. chinensis* A and B class genes with their homologs in other basal species and nature selection and protein interactions of B class MADS-box proteins were performed.

The differential regulation of *S. chinensis* A and B class MADS-box transcription factors likely plays a key role during bracts development. This study contributes to our understanding of bracts development in *Saururus*, an understudied paleoherb lineage.

## Results

### Physiological and morphological differences between green and white bracts

The stomatal conductance (gs) of white bracts was greatly reduced (Fig 1A). The net photosynthetic rate of white bracts was significantly (44%) lower than that of young green bracts in this species [17]. The abundance of chlorophyll a, chlorophyll b and carotenoids were all also

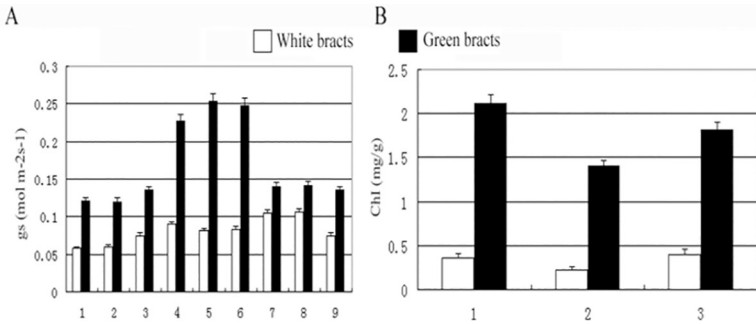

**Fig 1. Stomatal conductance (gs) and chlorophyll concentrations of the green bracts and white bracts of *S. chinensis*.** Stomatal conductance (gs) and chlorophyll concentrations of white and green bracts from different plant leaves were measured, respectively. Each sample was done in triplicate, and the error bars represent the standard deviation of the three independent experiments.

dramatically reduced in the white bracts of *S. chinensis* as compared with their levels in the green bracts (Fig 1B).

Morphological observation of the epidermis of green bracts found that these tissues completely lacked cone-shaped cells on either adaxial (Fig 2A) or abaxial surfaces (Fig 2B and 2C). The surfaces of white bracts, conversely, were found to display tube trichomes and the ridged, cone-shape epidermal cells common to petals, both on the adaxial (Fig 2G and 2H) and abaxial sides (Fig 2D–2F and 2I). Electron micrographs of ultrathin sections from both types of bracts showed that chloroplasts from normal green bracts had well-developed membrane systems composed of grana connected by stroma lamellae (Fig 3A and 3B). However, the thylakoid membrane systems in the petaloid bracts chloroplasts were disturbed, and the membrane spacing was not as clear as that in the normal green bracts (Fig 3D and 3E). The chloroplasts from petaloid bracts were smaller (Fig 3D and 3E) than those from green bracts (Fig 3A and 3B) and had a smaller average number of discs per grana stack (Fig 3D and 3E). Petaloid bracts also showed thicker cell walls (Fig 3F) than did green bracts (Fig 3C).

### Illumina sequencing and de novo assembly of sequence reads in *S. chinensis*

Clean reads from all samples were pooled and assembled de novo using the Trinity program [23], resulting in 86,532,094 total reads (10,816,511,750 bps) with an average length of 908bp in unigenes, an N50-value of 1582 bp, and with 13,821 unigenes (30.71%) longer than 1000 bp (Table 1). The 44,998 total unigenes were aligned to the four public protein databases (Nr, Swiss-Prot, KEGG, and COG), of which 22,668 (50.42%) were annotated in the public databases (S1 Table). Of these, 3,976 unigenes were found in all four databases, 22,617 (50.26%) were annotated in Nr databases, 16,772 (37.23%) in Swiss-Prot databases, 8,076 (17.94%) unigenes in COG databases, and 6,243 (13.43%) unigenes in KEGG databases (S1 Table). Furthermore, top-hit species distribution analysis based on BLASTx results showed the unigenes showing high sequence similarity with sequences of *Vitis vinifera* (5,904), *Theobroma cacao* (3,948), *Oryza sativa Japonica* Group (1,547), *Cucumis sativus* (1,326), *Fragaria vesca* subsp. *Vesca* (1154), *Glycine max* (974), *Solanum lycopersicum* (919), *Arabidopsis thaliana* (835), *Cicer arietinum* (804) and *Medicago truncatula* (737) (S2 Table). This suggests the *S. chinensis* genome is more closely related to *V. vinifera* genome than to other sequenced plant genomes.

### Gene Ontology (GO) classification

GO assignments were used to classify the functions of all unigenes. Based on sequence similarity, 22,688 unigenes were assigned to one or more ontologies. In total, 22,947 unigenes were

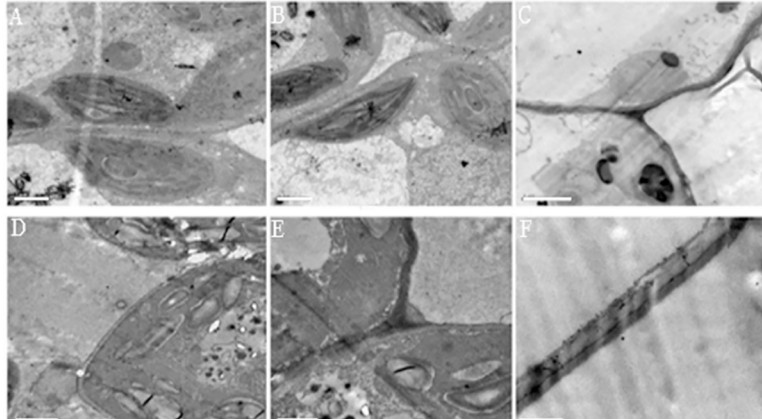

**Fig 2. Scanning electron microscope micrographs of green bracts and white bracts.** A) Adaxial surface of green bract. B, C) Abaxial surface of green bract. D, E, F, I) Abaxial surface of white bracts. G, H) Adaxial surface of white bract. Scale bars for A, C, D and G equal 100 μm, and scale bars for B, E, F, H and I equal 10 μm.

**Fig 3. Transmission electron micrographs of chloroplasts and cell walls of green bracts and white bracts of *S. chinensis*.** A, B) Chloroplast in green bract; D, E) Chloroplast in white bract; C) Cell wall in green bract; F) Cell wall in white bract; A and B, Scale bars represent 1 um; C, D, E and F, Scale bars represent 2 μm.

Table 1. Summary of illumina paired-end sequencing and assembly for *S. chinensis*.

| Database | Number | Total Length (bp) |
|---|---|---|
| Total Clean Reads | 86,532,094 | 10,816,511,750 |
| Q20 percentage | 95.49% | |
| GC percentage | 50.26% | |
| Average length of contigs | 16772 | |
| Number of unigenes | 44998 | 40,849,381 |
| Average length of unigenes | 907.8 | |
| Max length of unigenes | 15660 | |
| Min length of unigenes | 201 | |
| Unigene size N50 | 1582 | |

grouped under biological processes, 19,078 unigenes were grouped as cellular components, and 20,072 unigenes were grouped as having a role in molecular functions. In the biological processes category, the top GO terms included "metabolic process", "single-organism process", "localization", "establishment of localization" and "multicellular organismal process" (Fig 4). The top GO terms of the cellular component category included "cell", "organelle", "membran", "macromolecular complex" and "membrane-enclosed lumen" (Fig 4). Finally, the top terms in the molecular function category were "metabolic process", "binding", "single-organism process", "establishment of localization" and "biological regulation".

## Differentially Expressed Genes (DEGs) were identified in the petaloid and green bracts

A total of 43,463 genes were shown to be differentially expressed between green and white bracts, of which 5,797 were significant differential expression. Of these, 1,770 DEGs were up-

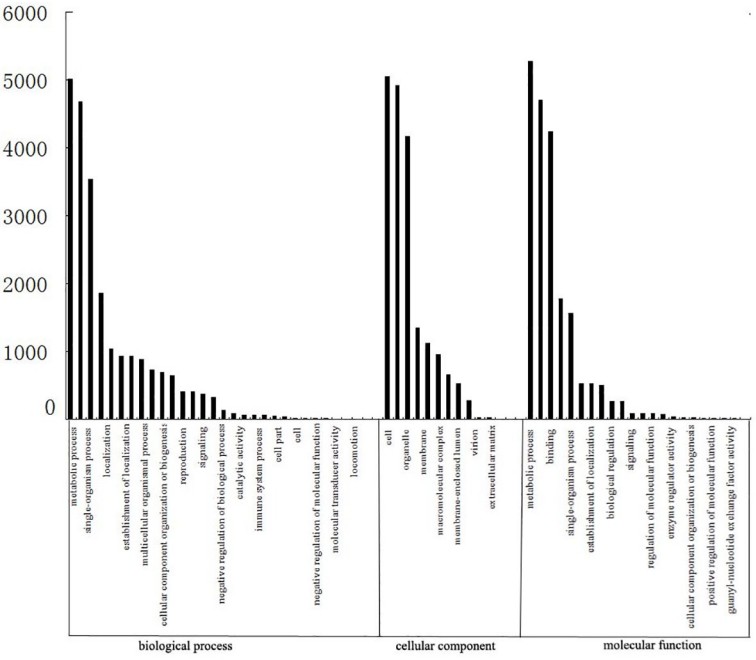

**Fig 4. Functional annotation of assembled unigenes based on Gene Ontology (GO) categorization.** The results are summarized in three main categories: biological process, cellular component and molecular function.

regulated and 4,027 DEGs were down-regulated in the green vs. white comparison (S3 Table). To understand the functions of these differentially expressed genes, we mapped them to terms in KEGG database, with special attention given to significantly enriched DEGs predicted to operate in metabolic or signal transduction pathways. Among all the genes with KEGG pathway annotation, 1,305 DEGs were annotated into 113 KEGG pathways in the green vs. white comparison (S4 Table). Among these, the most DEGs were found in the following pathways: "ribosome", "DNA replication", "ribosome biogenesis in eukaryotes", "mismatch repair", and "homologous recombination" (S4 Table).

To further compare the transcript abundance of DEGs in target tissues, a custom cDNA microarray was designed and hybridized with tissue-specific transcripts. Total RNA was isolated from young seedling leaves, as well as from bracts at various stages of development. That is, during the transition of green bracts to white bracts when the organ is half green and half white, we isolated the green tissue and the white tissue and extracted total RNA from each. Likewise, during the white bracts' reversion to a pale green coloration following floral senescence, we isolated the tissues by color and extracted RNA from the different samples. Comparing the bract tissues' transcript profiles with the transcript profiles of developing seedling leaves identified 166 bract DEGs (S5 Table). During the green-to-white transition, 66% (71) were up-regulated and 34% (37) were down-regulated in the green portion of the bicolor bracts, 79% (81) were up-regulated and 21% (21) were down-regulated in the white portion of the bracts, compared with 62% (23) of the DEGs being up-regulated and 38% (14) down-regulated in completely white bracts. During the bract white-to-green reversion, 36% (21) were up-regulated and 64% (38) were down-regulated in the white portion, and 8% (6) were up-regulated and 92% (71) were down-regulated in the pale green portion, and 9% (6) were up-regulated and 91% (58) were down-regulated in the fully pale green bracts (Fig 5) (S5 Table).

During the green-to-white transition, the white portion DEGs in the category of transcription were the most numerous (16%), of which 94% were up-regulated followed by metabolism (11%), cellular transport (9%), and energy (9%), a majority of which were down-regulated (66.67%) (S5 Table). In the fully white bracts, the most abundant functional category for DEGs

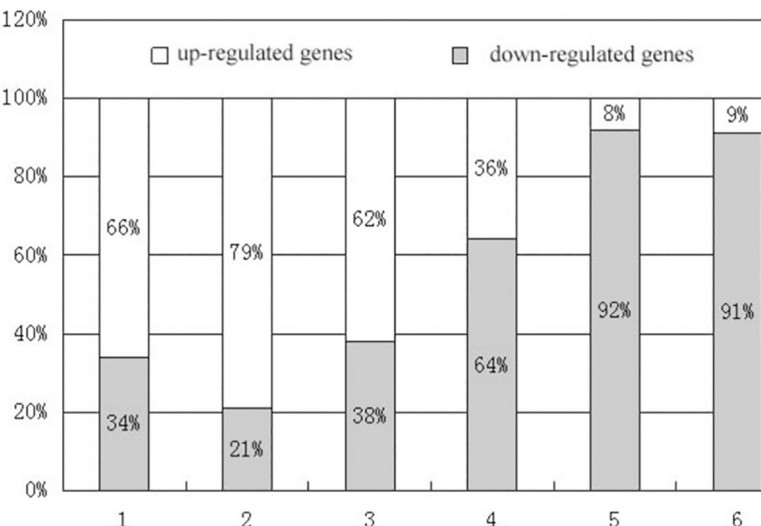

**Fig 5. Proportion of up-regulated (white) and down-regulated (grey) differentially expressed genes at different stages of bract development.** 1) Green tissue in bracts undergoing green-to-white transition; 2) White tissue in bracts undergoing green-to-white transition; 3) Fully white bracts; 4) White tissue in bracts during white-to-green reversion; 5) Pale green tissue during white-to-green reversion; 6) Fully pale green bracts.

was transcription (40.54%), for which all of the DEGs were up-regulated, followed by cellular transport (13.51%) and energy (10.81%), which were down-regulated (S5 Table). Conversely, in the white portions of bracts during the reversion to a pale green coloration following floral senescence, the category with the most DEGs was energy (15.79%), two-thirds of which were down-regulated, as well as transcription (14.04%) and cellular transport (14.04%), up to 75% of which were down-regulated. In the pale green portion of reverting bracts, transcription factors were most numerous (14.1%), 72.73% of which were down-regulated, followed by energy (11.54%), which were all down-regulated, cellular transport (7.69%), of which 83.33% were down-regulated, as well as binding (7.69%) and metabolism (6.1%) (all down-regulated). Among the DEGs of the fully pale green bracts, cellular transport (14.06%), transcription (12.5%), energy (10.94%) and metabolism (10.94%) were the most numerous and were all down-regulated except for the transcription factors. Not surprisingly, all putative genes encoding components of the chloroplast or involved in photosynthesis were down-regulated in the white bracts, including homologs to chlorophyll A-B binding protein (G136, G244 and G548) and a photosystem I reaction center subunit protein (G547) (S5 Table).

## Significantly up-regulated transcription factor homologs

Several transcription factor genes were found to be differentially expressed in the bracts compared to developing seedling leaves. The significantly up-regulated transcription factor genes included *ScAP1*, *ScSEP1*, *ScAGL6*, *ScPI* and *ScAP3*, in addition to other genes generically identified as MADS-box transcription factors and members of the zinc finger, basic helix-loop-helix (bHLH), MYB transcription factors and AUX/IAA protein families (S5 Table). The relative transcript abundances of these genes, however, differed between white and green bracts. For example, the zinc finger, basic helix-loop-helix (bHLH), and MYB transcription factors were up-regulated in white bracts and down-regulated in green bracts. The two *ScAP1* homologs were especially up-regulated in white bracts.

## Support of microarray results by real-time quantitative PCR

To evaluate the accuracy and reproducibility of the microarray results, ten differentially expressed transcription factors were selected for confirmation by qRT-PCR (Fig 6). The qRT-PCR results showed expression trends for these genes that were largely consistent with the microarray results. Among them, *ScAP1-A* (F1254) and *ScAP1-B* (GenBank accession *AY057378*, D.Z.L. unpublished data) MADS-box genes were more abundant in the fully white bracts and in the white portion of bracts whether during the green-to-white transition or during the white-to-green transition, in addition, *ScAP1-A* were more abundant in the green portion of bracts during the green-to-white transition (S5 Table) (Fig 6). *ScAP3-A* (EH662329) transcripts were more abundant in the green and white portion of bracts during the green-to-white transition, in the fully white bracts and in the white portion of bracts during the white-to-green transition, while *ScAP3-B* (EG530709) transcripts were more abundant in the white portion of bracts during the green-to-white transition and in the fully white bracts (S5 Table) (Fig 6). *ScPI-A* (EG530711) and *ScPI-B* (D658) transcripts were more abundant in the fully white bracts, meanwhile, *ScPI-B* (D658) transcripts were more abundant in the white portion of bracts during the green-to-white transition (S5 Table) (Fig 6).

## Alignment and phylogenetic analysis of *AP1* and B-class MADS-box transcription factors

Alignment with 24 putative AP1/FUL/FRU/FL/CAUL subfamily proteins and 5 putative MADS-box proteins showed that the *S. chinensis* putative *ScAP1-A* and *ScAP1-B* genes shared

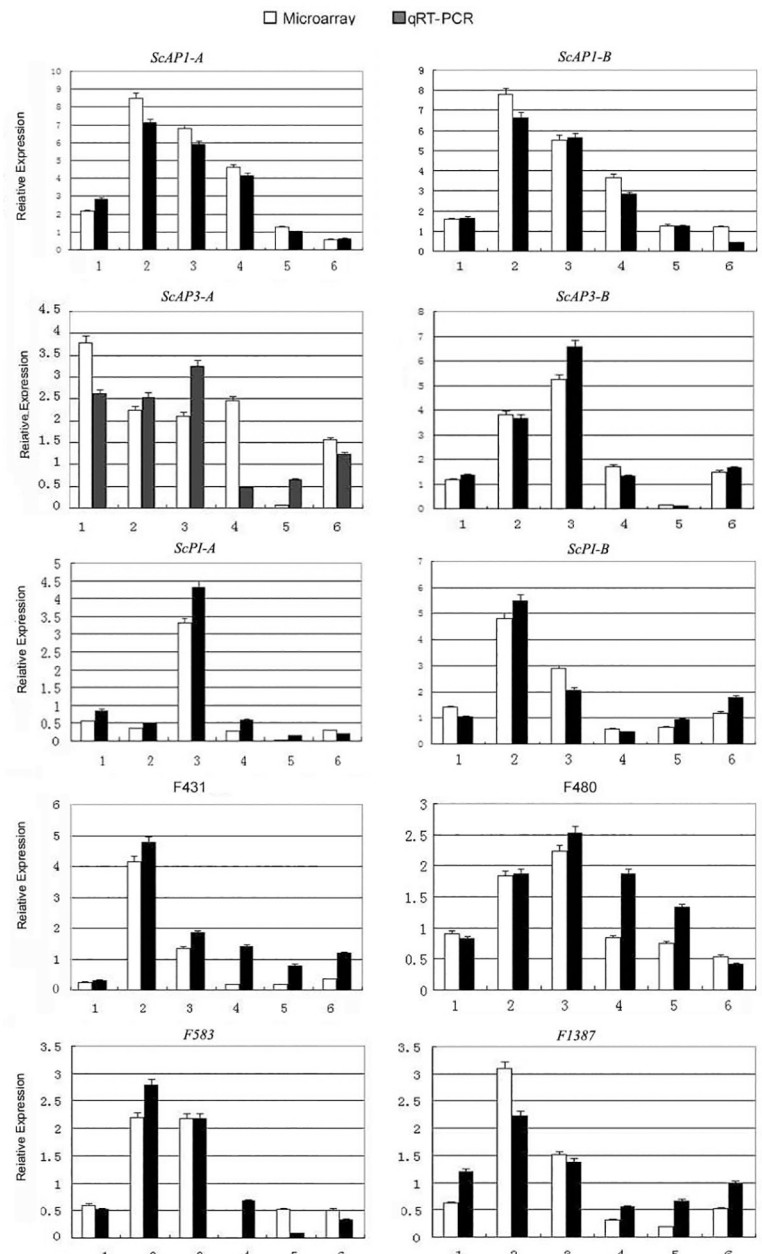

**Fig 6. Validation of microarray results by qRT-PCR for selected genes.** *ScAP1-A* and *ScAP1-B* MADS-box genes are F1254 and AY057378; *ScAP3-A* and *ScAP3-B* MADS-box genes are EH662329 and EG530709; *ScPI-A* and *ScPI-B* MADS-box genes are EG530711 and D658; the transcription factors are F431, F480, F583 and F1387. $\alpha$-*TUBULIN* (JK704891) was used as an internal control for normalization of the template cDNA. Each sample was done in triplicate, and the error bars represent the standard deviation of the three independent experiments. In each graph, the white bars represent the relative expression found in the microarray experiments, and the black bars represent the relative expression determined by normalized qRT-PCR. The number 1 to 6 correspond in the X axis represent the half green and half white during the transition of green to white bracts, the fully white, as well as the stages following floral senescence the half green and half white during the transition of white to green bracts, and Fully pale green bracts when bracts are reverting back to green coloration.

the MPPWML motifs of other *AP1* MADS-box genes (Fig 7A). 28 putative AP3/DEF subfamily proteins and 6 putative MADS-box proteins showed that the *S. chinensis* putative *ScAP3-A* and *ScAP3-B* genes shared the PI and paleoAP3 motifs of other *AP3* MADS-box genes

**Fig 7. Comparison of the C-terminal region of amino acid sequences for MADS-box B class genes.** ScAP1-A and ScAP1-B have a conserved MPPWML motif in the C-terminal region. ScAP3-A and ScAP3-B shared a PI-derived motif and a characteristic ancestral paleoAP3 motif. ScPI-A and ScPI-B protein has the highly conserved sequence PI-motif in the C-terminal region.

(Fig 7B), while 25 putative PI/GLO subfamily proteins and 3 other putative MADS-box proteins showed that the *S. chinensis* putative *ScPI-A* and *ScPI-B* genes shared the PI motifs of other PI MADS-box genes (Fig 7C).

Analysis of the A-class and B-class protein alignment using the Minimum Evolution method supported the identification of *S. chinensis AP1*, *AP3* and *PI*-family homologs. *ScAP1-A* and *ScAP1-B* were homologous to these *S. chinensis* A-class MADS-box genes, as all clustered together when compared to the AP1 homologs from *Houttuynia cordata*, *Magnolia* and *Persea americana*. In the phylogenetic reconstruction, *ScAP1-B* was placed close to an *AP1* homolog isolated from *H. cordata* with high bootstrap support (99%), and the pair clustered closely to *MADS600* homologs from *Asarum caudigerum* (Fig 8A). *ScAP3-A* and *ScAP3-B* were homologous to these *S. chinensis* B-class MADS-box genes, as all clustered together with moderate bootstrap support (BS = 86%) when compared to the *AP3* homologs from *Houttuynia cordata* and *Piper nigrum*. *ScAP3-A* was placed close to an *AP3* homolog isolated from *P. nigrum* with high bootstrap support (99%), and *ScAP3-B* was placed close to the *AP3* homolog isolated from *H. cordata* with high bootstrap support (100%) (Fig 8B). *ScPI-A* and *ScPI-B* were homologous to these *S. chinensis* B-class MADS-box genes, as all clustered together with high

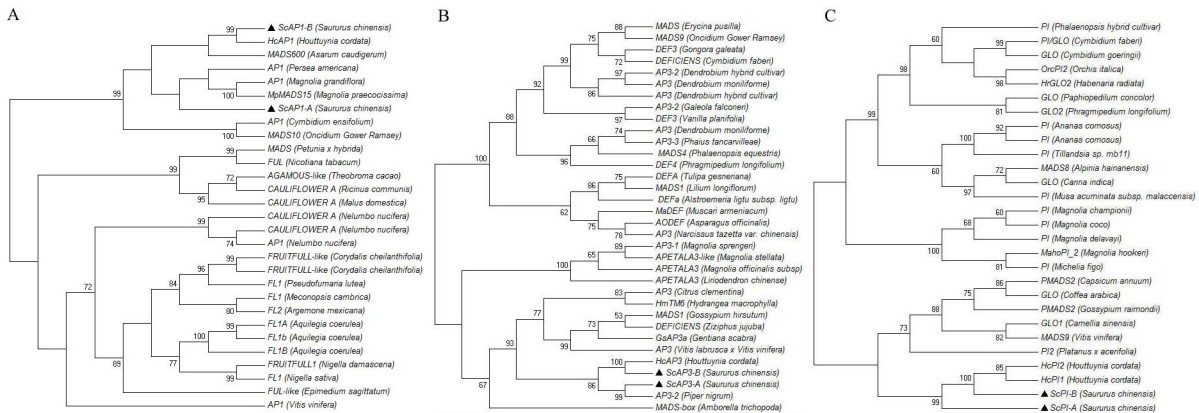

**Fig 8. Bootstrap consensus tree of Minimum Evolution method analysis of the putative A-class and B-class MADS-box transcription factors, using 1000 bootstrap iterations.** Black triangles represent the A-class and B-class MADS-box genes from *S. chinensis* in this study and others from GenBank that were included in the analysis.

**Table 2. Pairwise dN/dS analyses on five MADS-box genes with AP3 and nine MADS-box genes with PI.**

| AP3 | | | | | | | | |
|---|---|---|---|---|---|---|---|---|
| *ScAP3-A Saururus chinensis* | | | | | | | | |
| *ScAP3-B Saururus chinensis* | 0.4995 | | | | | | | |
| *HcAP3 Houttuynia cordata* | 0.3288 | 0.6745 | | | | | | |
| AP3 *Amborella* | 0.1691 | 0.3104 | 0.0880 | | | | | |
| AP3 *Magnolia* | 0.1141 | 0.2401 | 0.1892 | 0.1116 | | | | |
| PI | | | | | | | | |
| *ScPI-A Saururus chinensis* | | | | | | | | |
| *ScPI-B Saururus chinensis* | 0.3906 | | | | | | | |
| *HcPI1 Houttuynia cordata* | 0.3460 | 0.3343 | | | | | | |
| *HcPI2 Houttuynia cordata* | 0.3056 | 0.3977 | 0.8685 | | | | | |
| GLO *Paphiopedilum concolor* | 0.2030 | 0.1470 | 0.1758 | 0.1835 | | | | |
| PI *Magnolia championii* | 0.1568 | 0.2539 | 0.2543 | 0.2077 | -1.0000 | | | |
| PI *Magnolia delavayi* | 0.1768 | 0.2392 | 0.2488 | 0.2083 | 0.0440 | 0.3509 | | |
| MahoPI *Magnolia hookeri* | 0.1549 | 0.2331 | 0.2372 | 0.2049 | 1.0000 | 0.6617 | 0.4176 | |
| MADS9 *Vitis vinifera* | 0.1862 | 0.3516 | 0.3067 | 0.2937 | 0.1084 | 0.1061 | 0.1010 | 0.0960 |

bootstrap support (BS = 99%) when compared to the PI homologs from *H. cordata*. *ScAP3-B* was placed close to the *PI* homolog isolated from *H. cordata* with high bootstrap support (100%) (Fig 8C).

### Nature selection of B-class MADS-box transcription factors

The pairwise dN/dS analysis within *AP3* subfamily suggests that the tree display dN/dS values that are consistent with purifying selection. The ω values are not less than 1 among five *AP3* MADS-box genes and are constrained (dN/dS = 0.4995) between the two *ScAP3-A* and *ScAP3-B* MADS-box genes (Table 2). Meanwhile, the ω values are not less than 1 among nine PI MADS-box genes and are constrained (dN/dS = 0.3906) between the two *ScPI-A* and *ScPI-B* MADS-box genes (Table 2).

### Protein interactions of four B-class MADS-box proteins

To determine the protein interactions of the B class proteins in vitro, yeast two-hybrid assays were performed between the B class proteins. The results of two-hybrid analysis revealed that the ScPI-A protein could interact with ScPI-B. No homodimer or other interaction between the AP3-type members was observed in *S. chinensis* (Fig 9).

## Discussion

### Anatomical and photosynthetic physiological variation

In addition to the anatomical differences between the green and white bracts of *S. chinensis*, we have demonstrated in this study that bracts have reduced or altered foliar photosynthetic functions [17]. More specifically, in white bracts the fine structures of chloroplasts were degraded relative to those in normal leaves, and chlorophyll content, photosynthetic rate, and stomatal conductance (gs) were also greatly reduced (Figs 1, 3) [17]. These anatomical and physiological changes support the hypothesis that functional divergence has occurred in the white bracts, in a manner similar to the white bracts of *D. involucrate* [16].

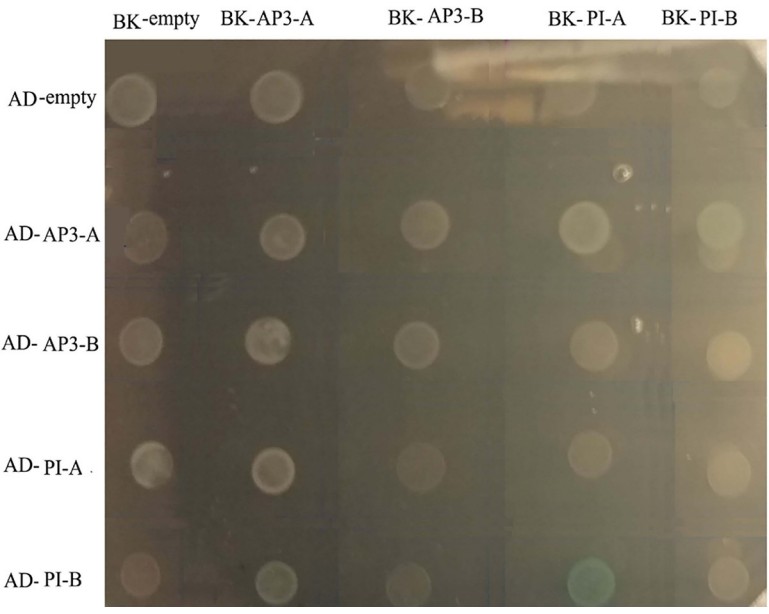

**Fig 9. Yeast two-hybrid interactions of five MADS-box B class proteins of *S. chinensis*.** GAL4-based yeast two-hybrid vectors pGADT7 vector (AD) and pGBKT7 vector (BK) and co-transformed into the AH109 yeast strain. Yeast growth on selection plates SD/-Trp-Leu-His-Ade/X-α-GAL solid medium.

## Differential gene expression in the green vs. white bracts

A total of 43,463 genes were shown to be differentially expressed between green bracts and white bracts, of which 5,797 were up-regulated or down-regulated. In the developmental transition of green bracts to white bracts, the majority of DEGs were up-regulated compared with the seedling leaves, while the majority of DEGs were down-regulated during the bracts' reversion to pale green coloration (Fig 5). Many of the DEGs categorized as functioning in transcription and cellular transport were up-regulated during the bracts' green-to-white transition, and down-regulated during the white-to-green reversion (S5 Table). DEGs identified as involved with energy and photosynthesis were down-regulated in white bracts (S5 Table), and their low or absent expression levels were confirmed by qRT-PCR analyses (Fig 6). Moreover, the changes in gene expression occurred before morphological characteristics appeared, as demonstrated by large number of genes that were up-regulated in green portion of the bracts during the green-to-white transition (S5 Table).

## The combined gene expression of *ScAP1*, *ScAP3* and *ScPI* paralogs plays an important role in the petaloid bracts

*AP1* is an A-class gene involved in the earliest stages of floral meristem development, and specifies petal and sepal identities in the first and second floral whorls; in *Arabidopsis*, loss-of-function mutants showed loss of petals and production of bract-like organs in lieu of sepals, among other defects [24]. According to both the microarray and qRT-PCR analyses, the two *AP1* homolog (*ScAP1-A* and *ScAP1-B*) were up-regulated in *S. chinensis* bracts during the green-to-white transition. Interestingly, *ScAP1-A* and *ScAP1-B* showed maximum up-regulation in the white portions of transitioning bracts, and then were less up-regulated in fully white bracts. Following floral senescence, the *ScAP1* homologs showed gradual down-regulation in bracts reverting to pale green color (S5 Table) (Fig 6). The likely involvement of these A-class genes

in executing bract petaloidy development appears to be another instance of a key floral development gene gaining expanded roles [5].

The products of MADS-box genes *AP3* and *PI* generally dimerize and specify floral organ petaloidy [5]. In this study, two paralogs of *AP3* and two paralogs of *PI* were isolated from the basal angiosperm *S. chinensis*, suggesting two gene duplication events for each gene lineage in this species. *ScAP3* transcripts were more abundant in the white portion of bracts during the green-to-white transition and in the fully white bracts (S5 Table) (Fig 6). *ScPI-A* transcripts were more abundant in the white portion of bracts during the green-to-white transition, while *ScPI-B* transcripts were more abundant in the fully white bracts (S5 Table, Fig 6). This may reflect an increased role for one of the paralogs in a certain tissue type. *ScAP3-B* transcripts were more abundant in the bracts during the green-to-white transition (S5 Table).

In many perianthless species, first and second whorl primordia initiate but then arrest; in Saururaceae and Piperaceae, conversely, petal and sepal primordia are completely absent throughout floral morphogenesis [25]. As such, it has been shown that selection pressures ($\omega < 1$) on *AP3* and *PI* would be reduced in the perianthless Piperales (Table 2), as these genes have been released from their perianth identity function and the same should be true for Sauruaceae [26]. We therefore suggest that two factors, the reduced dimerization capabilities of the various *S. chinensis* AP3/PI combinations and the complete lack of perianth tissues in this species (Fig 9), combined to render *S. chinensis* B-class genes especially free from evolutionary pressures. It will be interesting to determine what functions the broad expression patterns of the *AP3* and *PI* genes have maintained in this perianthless species.

Bracts that appear in a whorl subtending an inflorescence are collectively called an involucre. They often contain chloroplasts and can contribute to the photosynthetic capacity of the plant [27]. An involucre is a common feature beneath the inflorescences of many species, including *Euphorbia pulcherrima*, *Davidia* and the fossil genus *Amersinia*, all of which have large colorful bracts surrounding much smaller, less colorful flowers. The showy bracts of such species function to supplement or substitute absent or reduced perianths. In those species, large bracts developed to aid in attracting pollinators [17]. This likely occurred before the divergence of bougainvillea, *Euphorbia pulcherrima*, *Davidia* and *Amersinia* [28]. Because *Amersinia* also had a weakly developed perianth, the bracts might have appeared petaloid, as in *S. chinensis*. It has been suggested that gene networks controlling color and other petaloid characteristics in petals or tepals are ectopically expressed in such showy bracts [5].

In conclusion, the results indicated that petaloid bracts display anatomical and gene expression features normally associated with petals, as has been found in petaloid bracts of other species, supporting an evolutionarily conserved developmental program for petaloid bracts. This work has also demonstrated that A- and B-class gene expression are involved in the development of showy bracts to attract pollinators.

## Methods

### Plant material

An inbreeding population of *S. chinensis*, a perennial medicinal herb in the Saururaceae, was cultivated in the Botanical Garden of the Kunming Institute of Botany, Chinese Academy of Sciences, Kunming, Yunnan Province, China. The species is found mainly in moist habitats across southern China [29]. The bracts of this species are initially small and green (Fig 10A), resembling leaves, but white coloration develops first along the midvein and then spreads outward during inflorescence development (Fig 10B). Bracts then revert to a pale green coloration during floral senescence. There is some irregularity in bract morphology and color pattern, e.g., occasionally a plant's upper-most bracts remain very small with white coloration on both

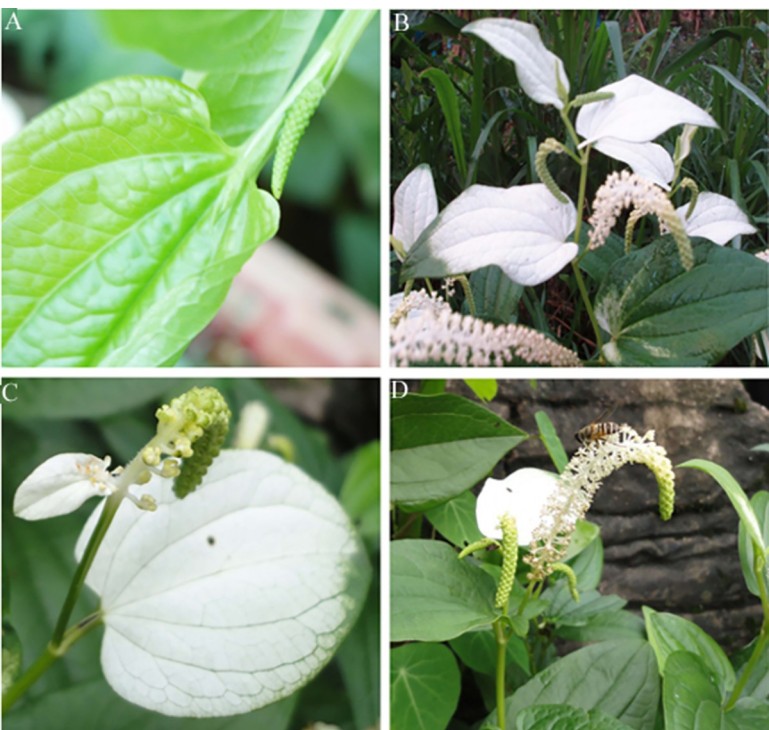

**Fig 10. Morphological diversity of the green bracts and white bracts of *Saururus chinensis*.** A) Bracts are initially green. B) Top three bracts are white. C) Occasionally, upper-most bracts are reduced in size and colored white on both adaxial and abaxial surfaces. D) A bee feeding from flowers of *S. chinensis*.

adaxial and abaxial surfaces (Fig 10C). More often, however, the three uppermost bracts are quite large (Fig 10B). The species with perianthless flowers tend to be wind pollinated [17] (Fig 10D).

## Scanning Electron Microscopy (SEM) and Transmission Electron Microscope (TEM) observation

Green bracts and white, petaloid bracts were dissected in the greenhouse and fixed in FAA (5% formaldehyde, 5% acetic acid, in 70% alcohol) for 13 hours, and then dehydrated through a series of alcohol solutions ranging from 70% to 100%. The materials were further dissected under a stereomicroscope, and the alcohol replaced with isopentanol acetate before the samples were dried in a Hitachi HCP-2 CO2 Critical Point Dryer (CPD). The dried material was mounted on stubs and coated with gold-palladium. Observations were made using a Hitachi KY Amray-100B SEM at 25KV.

The leaves were cut into 0.5mm × 0.5mm pieces and immediately fixed in 4% glutaraldehyde (in phosphate buffer, pH 7.4) for 3 hours in a refrigerator, then washed three times with the same buffer, each for 30 min. The specimens were re-fixed with 1% osmic acid (in the same buffer) overnight in a refrigerator, washed three times with the buffer and distilled water (30 min each time), and then were dehydrated in an alcohol series and embedded in Epon 812 resin. The specimens were sectioned and stained with uranium acetate-lead citrate, before being examined and photographed under a JEM-100 CX transmission electron.

## Stomatal conductance (gs) and chlorophyll concentrations measurement

All photosynthetic measurements were taken at this site between 16 May and 26 July, in both 2012 and 2013. Chlorophyll and carotenoid concentrations were determined using a modified

version of the spectrophotometric method of Hendry and Grime [30] and was done in triplicate. Leaf samples (0.2 g fresh weight) were homogenized in 3 ml 80% acetone (80% (v/v)). Samples were placed in darkness for 30 min, and then the absorbance of the supernatant was measured at 480, 645 and 663 nm to determine carotenoid and chlorophyll concentrations. Each sample was done in triplicate, and the mean and standard deviation of the three independent experiments were calculated.

## Tissue collection and RNA extraction

Bracts were collected at various stages of their development from floral meristem emergence to anthesis, including the initial green bract stage, the half green/half white stage, the fully white stage, as well as the stages following floral senescence when bracts revert back to green coloration. Collected issues were immediately frozen in liquid nitrogen and stored at -80˚C. Young upper leaves were also collected at the seedling stage and frozen. Total RNA was isolated from the tissues using TRIZOL reagent (Shanghai Huashun Co. Ltd., Shanghai, China) according to the manufacturer's protocol. RNA quality was characterized initially on an agarose gel and a NanoDrop ND1000 spectrophotometer (NanoDrop Technologies, Wilmington, DE, USA) and then further assessed by RIN (RNA Integrity Number) value (>8.0) using an Agilent 2100 Bioanalyzer (Santa Clara, CA, USA).

## Library preparation and sequencing

According to the Illumina manufacturer's instructions, poly(A)+ RNA was purified from 5 ug of pooled total RNA from petaloid-bract and green-bract using oligo(dT) magnetic beads. Fragmentation buffer was added to disrupt mRNA to short fragments of about 150–200 bp in size. Using these short fragments as templates, a random hexamer primer was used to synthesize the first-strand cDNA. Next, 10×buffer, 25 mM dNTPs, 20–60 U/μl RNaseH and 5 U/μl DNA polymerase I were added to synthesize the second strand. The double strand cDNA was then purified with QiaQuick PCR extraction kit (Qiagen) and washed with EB buffer for end repair and adenine addition. Finally, sequencing adaptors were ligated to the fragments. The fragments were purified by agarose gel electrophoresis and enriched by PCR amplification to construct a cDNA library. The library products were ready for sequencing analysis via Illumina HiSeq™ 2000.

## Sequence data analysis, assembly, and annotation

The cDNA library was sequenced on the Illumina sequencing platform and the sequencing method was pair end. Clean reads were obtained by removing from the raw data all low quality reads as well as those reads containing poly-N or adapter sequences. Subsequently, Q20, GC content and sequence duplication levels of the clean data were calculated. Unigenes were used for BLAST searches with annotation against the NCBI Nr database (NCBI non-redundant sequence database) using an E-value cut-off of $10^{-5}$ (E-value <0.00001). After sequence assembly, the unigene sequences were also aligned by BLASTX to protein databases such as Swiss-Prot, KEGG and COG.

## Screening of differentially expressed genes

Unigenes that showed different expression levels between the petaloid-bract and green-bract were subjected to GO function analysis and KEGG pathway analysis. A strict algorithm was developed to identify differentially expressed genes between two samples, building on previous work into the significance of digital gene expression profiles [31]. FDR (false discovery rate)

was used to determine the threshold P-value, and fold changes for each unigene between sample pairs (Green bracts vs. Showy bracts) were computed as the log2Ratio values. This research used FDR < 0.001 and an absolute value of the log2Ratio ≥ 1 as the threshold to judge the significance of gene expression difference [32]. For pathway enrichment analysis, all differentially expressed genes were mapped to terms in the KEGG database, and a list of KEGG terms that were significantly enriched in the bract phases compared to the genome background was compiled.

### Gene annotation

Functional categories of the predicted genes were obtained by applying gene ontology (GO) terms (http://www.geneontology.org) to the Nr database annotation using the Blast2GO program [33], and summarized using WEGO software [34]. Subsequently, the GO annotations of the differentially expressed genes were mapped to the plant-specific GO slim ontology using the map2slim script (www.geneontology.org/GO.slims.shtm l) (p-value, 0.05), and final classification of the differentially expressed genes was based on these GO slims.

### cDNA microarray construction and analysis

Reverse-subtracted white bract cDNA libraries were used to design a custom microarray (Hua Da Genomic Company, Beijing, China). Microarray probes were derived from the 1,262 uni-ESTs according to Zhang et al. [35] with minor modifications. The probes were generated by reverse transcription of 50 µg of total RNA using the SuperScript Indirect cDNA Labeling System (Invitrogen, Carlsbad, CA, USA). The reverse transcription reaction also contained 1 ng of Lambda polyA+ RNA-A (TX802; Takara, Kyoto, Japan) as an external control. Lambda DNA (TX803; Takara, Kyoto, Japan) was used as a control template, and SARS virus genes and distilled water were used as negative controls. Probes derived from RNA of seedling leaves were labeled with Cy3 fluorescent dye, and probes derived from RNA from bracts at various stages of flowering development were labeled with Cy5 fluorescent dye. Equivalent quantities of Cy3- and Cy5-labeled probes were pooled, purified, dried under vacuum according to Zhang et al. [35], and resuspended in 1× hybridization solution (5× standard saline citrate, 0.1% SDS, 25% formamide, and 0.1 mg/mL of denatured salmon sperm DNA). A common reference design was adopted in our cDNA microarray experiment [36], and two biological replicates for each time point were completed. Pre-hybridization, hybridization, and post-hybridization washes were carried out in accordance with the UltraGAPS Coated Slides instruction manual (Corning, Lowell, MA, USA). After being washed, the slides were scanned using an Axon GenePix 4100B scanner (Molecular Devices, Sunnyvale, CA, USA) at a resolution of 10 µm. Laser and photomultiplier tube voltages were adjusted manually to minimize background to reduce the number of spots that showed saturated signal values, and to bring the signal ratio of the majority of control genes as close to 1.0 as possible. Spot intensities were quantified using Axon GenePix pro 6.0 software (Molecular Devices, Sunnyvale, CA, USA). The intensity of each channel for individual spots was calculated by determining the median value. Background fluorescence was calculated on the basis of the fluorescence signal of the negative control genes. The Lambda control template DNA fragment was used to normalize the fluorescence intensities of the two labeling dyes for dye bias reduction. The spots with a ratio ≥2 or ≤0.5 were judged significantly upregulated and analyzed with Hierarchical Clustering Explorer 3.0 (http://www.cs.umd.edu/hcil/hce/hce3.html). Only those genes with ≥2-fold or ≤0.5-fold differences in expression were considered differentially expressed genes.

## Real-time quantitative PCR analysis

Total RNA of the seedling leaves and the bracts at various stages of development were reverse transcribed with oligo-dT and reverse transcriptase (Promega, Madison, WI, USA) following the supplier's protocols. To validate the cDNA microarray results for ten putative transcription factors including *ScAP1*, *ScAP3* and *ScPI*, real-time quantitative PCR (qRT-PCR) was performed using an ABI Prism 7500 HT sequence detection system (Applied Biosystems, Carlsbad, CA, USA). Primers were designed using Primer5 and synthesized in ShengGong Cooperation (Shanghai, China) (S6 Table). An alpha-tubulin gene sequence (JK704891) was used as internal control for normalization of the template cDNA. qRT-PCR reactions used 1× SYBR Green I PCR Master Mix (Applied Biosystems, Carlsbad, CA, USA) containing 200 nM of each primer and 1μl 1:10 diluted total cDNA. The PCR was performed using the following program: 1 cycle at 50˚C for 2 min, 1 cycle at 95˚C for 10 min, and then 40 cycles of 95˚C for 30 s, 60˚C for 30 s, and 72˚C for 18 s. Following amplification, melting curve analyses were performed at 95˚C for 10 min, followed by 40 cycles of 95˚C for 30 s, 60˚C for 30 s, and 72˚C for 20 s. The data collected during each extension phase were analyzed using SDS2.1 (Applied Biosystems, Carlsbad, CA, USA). The transcript abundance of the eight putative transcription factors was calculated using the relative $2^{-\Delta\Delta CT}$ analytical method [37]. Each sample was done in triplicate, and the mean and standard deviation of the three independent experiments were calculated.

## Construction of phylogenetic trees and estimation of the dN/dS ratio

The putative *S. chinensis* homologs of the MADS-box genes *AP1*, *AP3* and *PI* were aligned with those closely related to A-class genes from GenBank using Clustal X [38], followed by manual adjustment where necessary. Alignments were created using MIKC domains. A Minimum Evolution method (ME) tree was constructed using the pairwise deletion option in MEGA7 [39]. Genetic distances were estimated under Close-Neighbor-Interchange model.

The codon alignment of cDNA was performed and the phylogenic tree was constructed with the MEGA7 using Maximum Likelihood method [39]. The nonsynonymous/synonymous substitution rate ratio ($\omega$ = dN/dS) was estimated by using the software PAML4.9e [40]. Site-specific methods were applied to calculate selection pressure.

## Yeast two-hybrid assays

The full-length coding cDNA sequences of *ScAP3-A*, *ScAP3-B*, *ScPI-A*, and *ScPI-B* genes containing restriction sites at the 5'and 3' ends were introduced into the GAL4-based yeast two-hybrid vectors pGBKT7 (Clontech) and pGADT7-Rec (Clontech) and co-transformed into the AH109 yeast strain by the LiAc/DNA/PEG method according to the Yeast Protocols Handbook from Clontech (http://www.clontech.com).

## Supporting information

**S1 Table. Summary of the annotation percentage of *S. chinensis* unigenes as compared to public databases.**
(XLS)

**S2 Table. Top-hit species distribution.**
(DOC)

**S3 Table. Genes showing significant differential expression between green bracts and white bracts.**
(XLS)

**S4 Table. Pathway of genes showing differential expression between green bracts and white bracts.**
(XLS)

**S5 Table. Putative identifications of *S. chinensis* genes, and their transcript abundance levels in white and green portions of bracts during the green-to-white transition, at the fully white stage, white and pale green portions of bracts during the white-to-green reversion, and at the fully pale green stage.**
(XLS)

**S6 Table. RT-qPCR primers.**
(XLS)

## Acknowledgments

We are very grateful to Zachary Larson-Rabin for his help with *Saururus chinensis* flower photographs and manuscript editing.

## Author Contributions

**Formal analysis:** Yin-He Zhao.

**Funding acquisition:** Yin-He Zhao, De-Zhu Li.

**Methodology:** Yin-He Zhao, Xue-Mei Zhang.

**Project administration:** De-Zhu Li.

**Resources:** Yin-He Zhao, Xue-Mei Zhang.

**Writing – original draft:** Yin-He Zhao.

**Writing – review & editing:** Yin-He Zhao.

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
