## [Decision Letter · Decision Letter 0]

10 Feb 2021

PONE-D-21-01094

Development of the petaloid bracts of a paleoherb species, Saururus chinensis

PLOS ONE

Dear Dr. zhao,

Thank you for submitting your manuscript to PLOS ONE. After careful consideration, we feel that it has merit but does not fully meet PLOS ONE’s publication criteria as it currently stands. Therefore, we invite you to submit a revised version of the manuscript that addresses the points raised during the review process.

The presentation and languague need serious improvements.

We look forward to receiving your revised manuscript.

Kind regards,

Yun Zheng, Ph.D

Academic Editor

PLOS ONE

2. We note that you are reporting an analysis of a microarray, next-generation sequencing, or deep sequencing data set. PLOS requires that authors comply with field-specific standards for preparation, recording, and deposition of data in repositories appropriate to their field. Please upload these data to a stable, public repository (such as ArrayExpress, Gene Expression Omnibus (GEO), DNA Data Bank of Japan (DDBJ), NCBI GenBank, NCBI Sequence Read Archive, or EMBL Nucleotide Sequence Database (ENA)). In your revised cover letter, please provide the relevant accession numbers that may be used to access these data. For a full list of recommended repositories, see http://journals.plos.org/plosone/s/data-availability#loc-omics or http://journals.plos.org/plosone/s/data-availability#loc-sequencing.

4. We noticed you have some minor occurrence of overlapping text with the following previous publication(s), which needs to be addressed:

https://journals.plos.org/plosone/article?id=10.1371/journal.pone.0053019

In your revision ensure you cite all your sources (including your own works), and quote or rephrase any duplicated text outside the methods section. Further consideration is dependent on these concerns being addressed.

Reviewers' comments:

Reviewer's Responses to Questions

**Comments to the Author**

1. Is the manuscript technically sound, and do the data support the conclusions?

Reviewer #1: Yes

Reviewer #2: Yes

2. Has the statistical analysis been performed appropriately and rigorously? 

Reviewer #1: Yes

Reviewer #2: Yes

3. Have the authors made all data underlying the findings in their manuscript fully available?

Reviewer #1: Yes

Reviewer #2: No

4. Is the manuscript presented in an intelligible fashion and written in standard English?

Reviewer #1: Yes

Reviewer #2: Yes

5. Review Comments to the Author

Reviewer #1: The authors compared the morphologies and physiological indices of petaloid and non-petaloid (green) bracts between the perianthless paleoherb S. chinensis in green and white ones. And compared the transcriptome in these two kinds of bracts, identified and verified some differential expressed genes, also discussed the possibility molecular mechanisms of petal development, which is new to us. However, I have some small questions or some mistakes to figure out in this manuscript in below:

1. In Abstract you mentioned the total differential expression genes(DEGs) were 5978, among them up-regulated DEGs were 1770 and down-regulated were 4770, that total among were 5977, not 5978, may this is a simple mistake by careless?

2. All the figures were not having a good quality, they were too blurred, could provide above 300 dpi pictures?

3. You mentioned the B class genes could be interact with each other, what the significance of these genes interaction with each other? And this work has also demonstrated that A- and B-class gene expression are involved in the development of showy bracts to attract pollinators, however, why don’t you detect the A-and B-class genes’ interaction?

4. You should improve your English more fluently for your manuscript.

Reviewer #2: Zhao et al. studied the development of petaloid bracts in Saururus chinensis. The topic is very interesting, and the results at the anatomy, physiology and gene expression levels support their conclusions. However, several places in result part were confusing, and need be clarified, and some places in Methods were even wrong. Here, I list some:

“RNA-seq abundance analysis identified 43,463 genes that were found to be differentially expressed in S. chinensis bracts. Of these, 5,978 showed strong differential expression, of which 1,770 were up-regulated and 4,207 down-regulated in green compared to white bracts”. That means that 90% genes identified in this study were differentially expressed. What is the criteria to design “differentially expressed gene” and “strong differential expression” used in this study?

“Clean reads from all samples were pooled together, and were further assembled de novo using the Trinity program [23], resulting in 86,532,094 total reads (10,816,511,750 bps) with an average length of 908bp, an N50-value of 1582 bp, andwith 13,821 unigenes (30.71%) longer than 1000 bp (Table 1). Your sequencing method (single end or pair end and length) need be clarified in Material and Methods, “average length of 908bp” is not the length of clean reads.

“22,688 unigenes were assigned to one or more ontologies. 6,727 unigenes were grouped under biological processes, 5,805 unigenes were grouped as cellular components, and 7,445 unigenes were grouped as having a role in molecular functions”. The total unigene number designed to three GO category is less than 22688, even though certain genes can be designed to 2 or 3 GO category.

“Comparing the bract tissues’ transcript profiles with the transcript profiles of developing seedling leaves identified 166 bract DEGs”. In the following sentence, authors showed the up/down regulated DEGs of different development stages compared with seedling leaves, what stage did the 166 DEGs specific to?

In Methods part:

“According to the Illumina manufacturer’s instructions, poly(A)+ RNA was purified from 5 mg of pooled total RNA from petaloid-bract and green-bract cDNA libraries using oligo(dT) magnetic beads”. 5 mg is misspelled or not? How polyA RNA can be isolated from cDNA libraries?

In “Screening of differentially expressed genes” part, “Unigenes that showed different expression levels between the two samples were subjected to GO function analysis and KEGG pathway analysis”, what is the two samples?

“Following amplification, melting curve analyses were performed at 95°C for 10 min, followed by 40 cycles of 95°C for 30 s, 60°C for 30 s, and 72°C for 20 s”, is this for melting curve analysis?

Other issues:

Labelling (A, B….) on figures are not clear.

Sequencing data should be deposited into databases for publicly available.

Authors should pay attention to the grammatical errors, such as “The wild is found mainly in moist sites in southern China from sea level to 1700 m [29]”.

6. PLOS authors have the option to publish the peer review history of their article (what does this mean?). If published, this will include your full peer review and any attached files.

Reviewer #1: No

Reviewer #2: No

---

## [Author Response · Author response to Decision Letter 0]

2 May 2021

Date: Feb 10 2021 01:58PM

To: "yinhe zhao" yhzhao808@163.com

From: "PLOS ONE" plosone@plos.org

Subject: PLOS ONE Decision: Revision required [PONE-D-21-01094]

Attachment(s): recomendations.docx

PONE-D-21-01094

Development of the petaloid bracts of a paleoherb species, Saururus chinensis

PLOS ONE

Dear Dr. zhao,

Thank you for submitting your manuscript to PLOS ONE. After careful consideration, we feel that it has merit but does not fully meet PLOS ONE’s publication criteria as it currently stands. Therefore, we invite you to submit a revised version of the manuscript that addresses the points raised during the review process.

The presentation and languague need serious improvements.

We look forward to receiving your revised manuscript.

Kind regards,

Yun Zheng, Ph.D

Academic Editor

PLOS ONE

2. We note that you are reporting an analysis of a microarray, next-generation sequencing, or deep sequencing data set. PLOS requires that authors comply with field-specific standards for preparation, recording, and deposition of data in repositories appropriate to their field. Please upload these data to a stable, public repository (such as ArrayExpress, Gene Expression Omnibus (GEO), DNA Data Bank of Japan (DDBJ), NCBI GenBank, NCBI Sequence Read Archive, or EMBL Nucleotide Sequence Database (ENA)). In your revised cover letter, please provide the relevant accession numbers that may be used to access these data. For a full list of recommended repositories, see http://journals.plos.org/plosone/s/data-availability#loc-omics or http://journals.plos.org/plosone/s/data-availability#loc-sequencing.

4. We noticed you have some minor occurrence of overlapping text with the following previous publication(s), which needs to be addressed:

https://journals.plos.org/plosone/article?id=10.1371/journal.pone.0053019

In your revision ensure you cite all your sources (including your own works), and quote or rephrase any duplicated text outside the methods section. Further consideration is dependent on these concerns being addressed.

Reviewers' comments:

Reviewer's Responses to Questions

Comments to the Author

1. Is the manuscript technically sound, and do the data support the conclusions?

Reviewer #1: Yes

Reviewer #2: Yes

2. Has the statistical analysis been performed appropriately and rigorously?

Reviewer #1: Yes

Reviewer #2: Yes

3. Have the authors made all data underlying the findings in their manuscript fully available?

Reviewer #1: Yes

Reviewer #2: No

4. Is the manuscript presented in an intelligible fashion and written in standard English?

Reviewer #1: Yes

Reviewer #2: Yes

5. Review Comments to the Author

Reviewer #1: The authors compared the morphologies and physiological indices of petaloid and non-petaloid (green) bracts between the perianthless paleoherb S. chinensis in green and white ones. And compared the transcriptome in these two kinds of bracts, identified and verified some differential expressed genes, also discussed the possibility molecular mechanisms of petal development, which is new to us. However, I have some small questions or some mistakes to figure out in this manuscript in below:

1. In Abstract you mentioned the total differential expression genes(DEGs) were 5978, among them up-regulated DEGs were 1770 and down-regulated were 4770, that total among were 5977, not 5978, may this is a simple mistake by careless?

2. All the figures were not having a good quality, they were too blurred, could provide above 300 dpi pictures?

3. You mentioned the B class genes could be interact with each other, what the significance of these genes interaction with each other? And this work has also demonstrated that A- and B-class gene expression are involved in the development of showy bracts to attract pollinators, however, why don’t you detect the A-and B-class genes’ interaction?

4. You should improve your English more fluently for your manuscript.

Reviewer #2: Zhao et al. studied the development of petaloid bracts in Saururus chinensis. The topic is very interesting, and the results at the anatomy, physiology and gene expression levels support their conclusions. However, several places in result part were confusing, and need be clarified, and some places in Methods were even wrong. Here, I list some:

“RNA-seq abundance analysis identified 43,463 genes that were found to be differentially expressed in S. chinensis bracts. Of these, 5,978 showed strong differential expression, of which 1,770 were up-regulated and 4,207 down-regulated in green compared to white bracts”. That means that 90% genes identified in this study were differentially expressed. What is the criteria to design “differentially expressed gene” and “strong differential expression” used in this study?

“Clean reads from all samples were pooled together, and were further assembled de novo using the Trinity program [23], resulting in 86,532,094 total reads (10,816,511,750 bps) with an average length of 908bp, an N50-value of 1582 bp, andwith 13,821 unigenes (30.71%) longer than 1000 bp (Table 1). Your sequencing method (single end or pair end and length) need be clarified in Material and Methods, “average length of 908bp” is not the length of clean reads.

“22,688 unigenes were assigned to one or more ontologies. 6,727 unigenes were grouped under biological processes, 5,805 unigenes were grouped as cellular components, and 7,445 unigenes were grouped as having a role in molecular functions”. The total unigene number designed to three GO category is less than 22688, even though certain genes can be designed to 2 or 3 GO category.

“Comparing the bract tissues’ transcript profiles with the transcript profiles of developing seedling leaves identified 166 bract DEGs”. In the following sentence, authors showed the up/down regulated DEGs of different development stages compared with seedling leaves, what stage did the 166 DEGs specific to?

In Methods part:

“According to the Illumina manufacturer’s instructions, poly(A)+ RNA was purified from 5 mg of pooled total RNA from petaloid-bract and green-bract cDNA libraries using oligo(dT) magnetic beads”. 5 mg is misspelled or not? How polyA RNA can be isolated from cDNA libraries?

In “Screening of differentially expressed genes” part, “Unigenes that showed different expression levels between the two samples were subjected to GO function analysis and KEGG pathway analysis”, what is the two samples?

“Following amplification, melting curve analyses were performed at 95°C for 10 min, followed by 40 cycles of 95°C for 30 s, 60°C for 30 s, and 72°C for 20 s”, is this for melting curve analysis?

Other issues:

Labelling (A, B….) on figures are not clear.

Sequencing data should be deposited into databases for publicly available.

Authors should pay attention to the grammatical errors, such as “The wild is found mainly in moist sites in southern China from sea level to 1700 m [29]”.

6. PLOS authors have the option to publish the peer review history of their article (what does this mean?). If published, this will include your full peer review and any attached files.

Do you want your identity to be public for this peer review? For information about this choice, including consent withdrawal, please see our Privacy Policy.

Reviewer #1: No

Reviewer #2: No

---

## [Decision Letter · Decision Letter 1]

2 Jun 2021

PONE-D-21-01094R1

Development of the petaloid bracts of a paleoherb species, Saururus chinensis

PLOS ONE

Dear Dr. zhao,

Thank you for submitting your manuscript to PLOS ONE. After careful consideration, we feel that it has merit but does not fully meet PLOS ONE’s publication criteria as it currently stands. Therefore, we invite you to submit a revised version of the manuscript that addresses the points raised during the review process.

i) The criteria used in the differential analysis needs to be clarified; ii) The comparisons of different data/results are expected; iii) The presentation should be serious improved.

We look forward to receiving your revised manuscript.

Kind regards,

Yun Zheng, Ph.D

Academic Editor

PLOS ONE

Journal Requirements:

Reviewers' comments:

Reviewer's Responses to Questions

**Comments to the Author**

1. If the authors have adequately addressed your comments raised in a previous round of review and you feel that this manuscript is now acceptable for publication, you may indicate that here to bypass the “Comments to the Author” section, enter your conflict of interest statement in the “Confidential to Editor” section, and submit your "Accept" recommendation.

Reviewer #1: All comments have been addressed

Reviewer #2: (No Response)

2. Is the manuscript technically sound, and do the data support the conclusions?

Reviewer #1: Yes

Reviewer #2: Yes

3. Has the statistical analysis been performed appropriately and rigorously? 

Reviewer #1: Yes

Reviewer #2: (No Response)

4. Have the authors made all data underlying the findings in their manuscript fully available?

Reviewer #1: Yes

Reviewer #2: Yes

5. Is the manuscript presented in an intelligible fashion and written in standard English?

Reviewer #1: Yes

Reviewer #2: Yes

6. Review Comments to the Author

Reviewer #1: This manuscript has been fully satisified my questions about what I am asking, and this manuscript has some new ideas about its topics, so I advise to accept it for publication on this journal.

Reviewer #2: In abstract and result parts, “RNA-seq abundance analysis identified 43,463 genes that were found to be differentially expressed in S. chinensis bracts. Of these, 5,978 showed strong differential expression, of which 1,770 were up-regulated and 4,207 down-regulated in green compared to white bracts”. You claimed “43,463 genes that were found to be differentially expressed in S. chinensis bracts’, then, you claimed “5,978 showed strong differential expression”, what is the criteria to design “differentially expressed gene” and “strong differential expression” used in this study?

Both RNA-Seq and microarray were used in this study, differentially expressed genes (DEGs) were screened with each method, how about the consistence between the DEGs of the two method?

“Scanning Electron Microscopy (SEM) and transmission electron microscope (TEM)” and other subtitles in method are not properly used.

7. PLOS authors have the option to publish the peer review history of their article (what does this mean?). If published, this will include your full peer review and any attached files.

Reviewer #1: **Yes: **Yang Jun

Reviewer #2: No

---

## [Author Response · Author response to Decision Letter 1]

19 Jun 2021

Reviewer #2:

1.In abstract and result parts, “RNA-seq abundance analysis identified 43,463 genes that were found to be differentially expressed in S. chinensis bracts. Of these, 5,978 showed strong differential expression, of which 1,770 were up-regulated and 4,207 down-regulated in green compared to white bracts”. You claimed “43,463 genes that were found to be differentially expressed in S. chinensis bracts’, then, you claimed “5,978 showed strong differential expression”, what is the criteria to design “differentially expressed gene” and “strong differential expression” used in this study?

We have revised in Manuscript.

The criteria to design “differentially expressed gene” and “strong differential expression” used in this study was that “differentially expressed gene” was different expression between green and white bracts, and the “strong differential expression” was FDR < 0.001 and an absolute value of the log2 ratio >1 as the threshold to determine the significant difference in gene expression. We thought the significant difference expression was strong differential expression in this study and have revised the strong differential expression to the significant difference expression in abstract and result parts.

2.Both RNA-Seq and microarray were used in this study, differentially expressed genes (DEGs) were screened with each method, how about the consistence between the DEGs of the two method?

The consistence between the DEGs of RNA-Seq and microarray was good.

3. “Scanning Electron Microscopy (SEM) and transmission electron microscope (TEM)” and other subtitles in method are not properly used.

Yes, we have revised in method of Scanning Electron Microscopy (SEM) and transmission electron microscope (TEM) observation and Stomatal conductance (gs) and Chlorophyll concentrations measurement.

---

## [Decision Letter · Decision Letter 2]

22 Jul 2021

Development of the petaloid bracts of a paleoherb species, Saururus chinensis

PONE-D-21-01094R2

Dear Dr. zhao,

We’re pleased to inform you that your manuscript has been judged scientifically suitable for publication and will be formally accepted for publication once it meets all outstanding technical requirements.

Kind regards,

Yun Zheng, Ph.D

Academic Editor

PLOS ONE

Additional Editor Comments (optional):

Reviewers' comments:

Reviewer's Responses to Questions

**Comments to the Author**

1. If the authors have adequately addressed your comments raised in a previous round of review and you feel that this manuscript is now acceptable for publication, you may indicate that here to bypass the “Comments to the Author” section, enter your conflict of interest statement in the “Confidential to Editor” section, and submit your "Accept" recommendation.

Reviewer #1: All comments have been addressed

Reviewer #2: All comments have been addressed

2. Is the manuscript technically sound, and do the data support the conclusions?

Reviewer #1: Yes

Reviewer #2: Yes

3. Has the statistical analysis been performed appropriately and rigorously? 

Reviewer #1: Yes

Reviewer #2: Yes

4. Have the authors made all data underlying the findings in their manuscript fully available?

Reviewer #1: Yes

Reviewer #2: Yes

5. Is the manuscript presented in an intelligible fashion and written in standard English?

Reviewer #1: Yes

Reviewer #2: Yes

6. Review Comments to the Author

Reviewer #1: I have no other comments about this manuscript,I think it could be published in this journal with a little revisement.

Reviewer #2: (No Response)

7. PLOS authors have the option to publish the peer review history of their article (what does this mean?). If published, this will include your full peer review and any attached files.

Reviewer #1: No

Reviewer #2: No

---

## [Editor Report · Acceptance letter]

25 Aug 2021

PONE-D-21-01094R2 

Development of the petaloid bracts of a paleoherb species, *Saururus chinensis*

Dear Dr. Zhao:

I'm pleased to inform you that your manuscript has been deemed suitable for publication in PLOS ONE. Congratulations! Your manuscript is now with our production department. 

Kind regards, 

on behalf of

Dr. Yun Zheng 

Academic Editor

PLOS ONE